# A Simple and Efficient Image Stabilization Method for Coastal Monitoring Video Systems

**Isaac Rodriguez-Padilla [1],\*** , **Bruno Castelle [1]** , **Vincent Marieu [1]** and **Denis Morichon [2]**

1   CNRS, UMR 5805 EPOC, Université de Bordeaux, 33615 Pessac, France; bruno.castelle@u-bordeaux.fr (B.C.); vincent.marieu@u-bordeaux.fr (V.M.)

2   SIAME-E2S, Université de Pau et des Pays de l'Adour, 64600 Anglet, France; denis.morichon@univ-pau.fr

\*   Correspondence: isaac.rodriguez-padilla.1@u-bordeaux.fr

**Abstract:** Fixed video camera systems are consistently prone to importune motions over time due to either thermal effects or mechanical factors. Even subtle displacements are mostly overlooked or ignored, although they can lead to large geo-rectification errors. This paper describes a simple and efficient method to stabilize an either continuous or sub-sampled image sequence based on feature matching and sub-pixel cross-correlation techniques. The method requires the presence and identification of different land-sub-image regions containing static recognizable features, such as corners or salient points, referred to as keypoints. A Canny edge detector (*CED*) is used to locate and extract the boundaries of the features. Keypoints are matched against themselves after computing their two-dimensional displacement with respect to a reference frame. Pairs of keypoints are subsequently used as control points to fit a geometric transformation in order to align the whole frame with the reference image. The stabilization method is applied to five years of daily images collected from a three-camera permanent video system located at Anglet Beach in southwestern France. Azimuth, tilt, and roll deviations are computed for each camera. The three cameras showed motions on a wide range of time scales, with a prominent annual signal in azimuth and tilt deviation. Camera movement amplitude reached up to 10 pixels in azimuth, 30 pixels in tilt, and 0.4° in roll, together with a quasi-steady counter-clockwise trend over the five-year time series. Moreover, camera viewing angle deviations were found to induce large rectification errors of up to 400 m at a distance of 2.5 km from the camera. The mean shoreline apparent position was also affected by an approximately 10–20 m bias during the 2013/2014 outstanding winter period. The stabilization semi-automatic method successfully corrects camera geometry for fixed video monitoring systems and is able to process at least 90% of the frames without user assistance. The use of the *CED* greatly improves the performance of the cross-correlation algorithm by making it more robust against contrast and brightness variations between frames. The method appears as a promising tool for other coastal imaging applications such as removal of undesired high-frequency movements of cameras equipped in unmanned aerial vehicles (UAVs).

**Keywords:** video monitoring; image stabilization; geo-rectification; shoreline change

## 1. Introduction

Over the past decades, the use of shore-based video systems has become a very popular and accessible low-cost tool for coastal monitoring, given their capability to deliver continuous, high-frequency (e.g., daily) data over large enough spatial scales [1,2]. Among the many applications using video-based remote sensing are shoreline and sandbar position tracking [3–5], nearshore bathymetry estimation [6–8], determination of intertidal beach slopes [9,10], rip channel formation and evolution, [11–14], estimation of longshore currents [15,16], wave celerity, period and direction [6,17,18], and breaking wave height [19].

Nevertheless, successful and reliable video-based products can only be produced if accurate image transformation into real-world coordinates is achieved. This is performed through photogrammetry techniques which provides relationships between the 2-D image geometry and the 3-D real-world coordinates. A common approach consists of using projective transformation [20,21] that usually takes into account two types of calibration: An intrinsic calibration, which accounts for the physical characteristics of the camera lens and can be obtained directly in the lab prior to field installation (in order to remove distortion effects), and an extrinsic calibration, which depends on the camera location and orientation after installation, as well as a set of surveyed ground control points (GCPs), correspondingly manually digitized from the image. Both calibrations are often done just once, assuming that the physical properties of the lens remain unchanged over time and that the video cameras and their mounting structures do not move. Hence, the real-world coordinates of the fixed GCPs are supposed to systematically coincide with the digitized GCPs' image coordinates for all video frames. However, the latter assumption is challenged by many observations, showing that even fixed-mounted video cameras are never perfectly static [1,22–24].

Common non-anthropogenic causes that may produce camera movement or vibration are attributed to thermal expansion and insulation effects, as well as rain and wind forcing [1,24]. This is particularly evident in outdoor installations where cameras are directly exposed to the elements. One possible but tedious solution to address this issue is to digitize all of the GCPs for each presumably displaced image in order to obtain separately geometric solutions [24]. Another potentially more efficient alternative is to use (semi-)automatic image stabilization.

Image stabilization refers to any technique used to create a video sequence in which any unwanted camera motion is removed from the original sequence. In coastal imagery, Holman and Stanley [1] were the first to address the problem of camera viewing angle variations due to thermal and wind effects. In order to compensate for the camera movement after acquisition, they used a template matching method that consisted of selecting small high-contrast regions of an image (including fixed objects and salient points) and match them with a reference image (from a survey date with a known geometry solution) to compute their displacement (deviations of tilt and azimuth) in terms of pixel shift. Using this technique, they found an almost daily diurnal signal with an amplitude of approximately 2 pixels in tilt as a result of the thermal response of the tower on which the cameras were mounted. After geo-rectification, this shift was equivalent to an error of approximately 30 m in the longshore direction 1000 m away from the camera. With the further increased use of Unmanned Aerial Vehicles (UAVs), this method has been refined and automatized to create and recognize features (virtual GCPss) within a user specified window and a brightness intensity threshold [25,26].

Similar automated feature-matching image stabilization methods have been developed using the same principle of image comparison. Pearre and Puleo [22] used correlation techniques, a procedure similar to particle image velocimetry (PIV; [27]) to correct camera azimuthal and tilt movement. They showed that geo-rectified image errors owing to camera movement grows nonlinearly with the distance between the camera and the object of interest. Therefore, even small pixel variations can lead to large errors in real-world coordinates (500 m error 2300 m away from the camera at Rehoboth Beach, Delaware; [22]). Moreover, they found that despite the fact that all of the cameras experienced slight movements, the motion effect after geo-rectification was more evident on the camera with the longest focal length, as this camera was more sensitive to small changes in tilt. Another semi-automatic image stabilization procedure is described by Vousdoukas et al. [23], Vousdoukas et al. [28] for UAVs and fixed cameras. Their method consists first of defining a mask to remove the sea from the land for all frames. The next step is to extract and match pairs of features (keypoints) between consecutive frames. This can be done by using either a Scale-Invariant Feature Transform (SIFT; [29,30]) or a Speeded-Up Robust Features (SURF) algorithm [31]. Both algorithms are capable of detecting and extracting distinctive features such as corners and salient points, under some variations of illumination, scale, and orientation between frames. In order to identify the correct matches, a random sample

consensus (RANSAC) algorithm [32] is implemented to filter the outliers, followed by a least squares fit approach, as well as a sub-sampling of keypoint pairs to reduce computation time.

All of the above techniques rely upon the presence of land-based features or any fixed region that includes objects with high contrast in the camera field of view. However, in some cases, cameras just look at the ocean covering only a small beach portion, offering few or no fixed salient features. Some studies have overcome this limitation by tracking the horizon and using it as a geometric constraint in order to estimate the camera position and stabilize the images [33,34]. On the other hand, anticipating or predicting the movement of a camera can also be considered as an alternative approximation to reduce image deviation after acquisition when no features are available in the camera view field. Bouvier et al. [24] identified the primary environmental parameters controlling the shift of the cameras (air temperature, wind conditions, nebulosity, and solar position) and used them to develop an empirical model to simulate camera movement. Their model showed a good agreement against the observed camera viewing angle deviations ($R^2$ = 0.71, $RMSE$ = 0.12° for tilt and roll anomaly) confirming that, at their site, camera motion was primarily controlled by thermal expansion of the poles where the cameras were mounted, which is modulated in magnitude by the solar azimuth angle and cloud coverage. Bouvier et al. [24] indicated that the rectification error is up to 200 and 20 m in the longshore and cross-shore direction, respectively, which can be decreased by 80% with their automated stabilization method.

Image stabilization is a fundamental topic in the research field of Computer Vision [35] with many applications in object recognition [36], motion analysis [37], and image restoration [38], among others [39]. Similar stabilization methods have been developed with other purposes outside the field of coastal imagery [40–43]. One primary problem with existing stabilization methods is that they are sensitive to changes in illumination, making them prone to failure [44]. Brightness and contrast variations due to illumination changes are typical for image sequences and can often change within a few frames [45]. Moreover, if the image sequence is sub-sampled, these parameters can substantially change between consecutive frames. The purpose of the present work is to develop a new straightforward method for image stabilization to overcome this limitation. This method provides a robust solution to stabilize an image sequence under varying illumination with the ability to process large databases. It builds on state-of-the-art techniques and available routines, with valuable input for the coastal imaging research community. The following sections will describe the basic steps to stabilizing an image sequence, compensating for camera motion to avoid positioning errors in the image. Furthermore, the image stabilization method will be applied to five years of continuous video data collected from a three-camera video system located at Anglet Beach in southwestern France.

## 2. Study Site and Video Data

Anglet Beach is a structurally-engineered embayed sandy beach located on the Basque Coast in southwestern France (Figure 1). The beach is delimited by a prominent headland in the South and extends over 4 km up to a 1 km long jetty protecting the entrance of the river mouth. According to the classification of Wright and Short [46], Anglet Beach is an intermediate-reflective beach composed of medium to coarse sand, with a steep beach-face slope (tan $\beta \approx 0.1$) characterized by a mostly double bar system [5,47]. Given the west-northwest (WNW) beach orientation, the coast is predominantly exposed to North Atlantic high-energy swells coming from the WNW direction with an average annual significant offshore wave height $H_s$ = 1.57 m (occasional wave heights >10 m during winter storms) and average peak period $T_p$ = 10 s [47,48]. Tides in the area are semi-diurnal with average ranges of 3.94 m for spring tides and 1.69 m for neap tides (i.e., meso-macro tidal range).

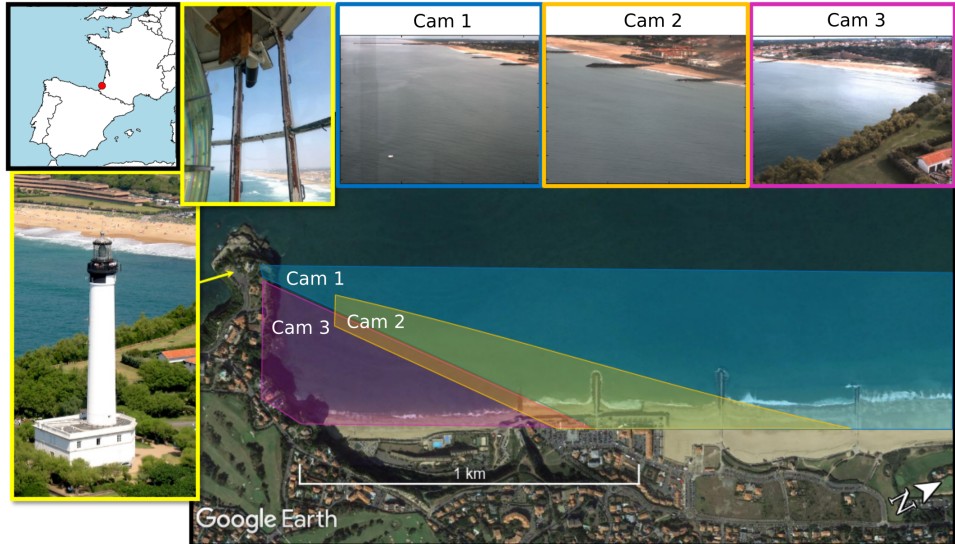

**Figure 1.** Map of Anglet Beach (Basque Coast, southwestern France) showing the location and field of view of the three-camera video system installed inside the Biarritz lighthouse (photo by M. Bourbon).

A permanent video-monitoring station has been operating since September 2013 at the southern end of Anglet Beach. It is installed inside the Biarritz lighthouse, 70 m above mean sea level. The station consists of three cameras: Camera 1 (12 mm lens), camera 2 (25 mm lens) and camera 3 (8 mm lens) that together cover approximately 2.5 km of the southern beach shoreline, including four groins that extend about 100 m seaward (Figure 1). Each camera provides different image products in the daylight. From September 2013 to February 2017, single snapshot images, 10 min averaged time-exposure images (timex), 10 min variance images, and 10 min timestack images (e.g., [1]) were collected every 20 min at 1 Hz. In March 2017, the configuration of the video station changed. From April 2017 to the present, image products were collected every 15 min at 1 Hz during an averaged period of sampling of 14-min. According to Castelle et al. [49], the best shoreline proxy for meso-macrotidal multiple-barred beaches was obtained at mean high water level (MHW), since the inner-bar and berm dynamics have little influence on the shoreline cross-shore displacement for this elevation contour. In order to have at least one image per day, the lowest high tide level was preferred. For this study, time-exposure images were selected at mean high water neap level time (MHWN = 0.86 m) with a mean accuracy of $\pm 0.08$ m and a maximum error of $\pm 0.28$ m. This resulted in a five-year daily time-exposure image time series consisting of approximately 1500 frames per camera, spanning from 1 October, 2013 to 9 September, 2018. The timex time series was treated as a continuous image sequence and was further used to compute the long-term image displacement for each camera.

## 3. Image Stabilization Method

The image stabilization method developed below is based on feature matching, and necessarily requires the presence of at least a few recognizable static features distributed in both dimensions in the field of view. Features should not be collinear, so that the geometric transformation can be correctly applied to the whole image. The concept of the method is to compute the two-dimensional displacement (azimuth and tilt) of the features between frames and use them as matching control points to estimate a 2-D geometric transformation that incorporates translation, rotation, and scaling to align the whole image with a reference image. The following subsections will describe the basic steps to stabilize an image sequence through a semi-automatic procedure using state-of-the-art techniques.

### 3.1. Reference Image and Sub-Image Region Selection

The first step consists of defining a reference image. All of the frame motions are estimated relative to this reference image. The selection of the reference image should correspond to the time

when the GCPs were surveyed. The next step is to identify static recognizable features within the reference frame. The features should be easily visible fixed objects such as buildings or any inland hard structure containing corners or salient points. The features should not only appear in the reference image, but also in every frame of the whole time series. Around each feature, a sub-image region (hereinafter referred to as zone) should be manually defined. This user-defined zone surrounding the feature must also contain a single reference point, referred to as keypoint, specified by the user. Every zone acts as a search region to identify the keypoint, representing the feature that would drifting with time. This implies that the user needs to estimate a possible range of camera pixel shift to design the zone within which the feature and keypoint is always visible. This can be done easily through a visual inspection of the entire image sequence. To test if the zone size is adequate, the zone from the reference frame can be compared with the last frame of the image sequence (assuming that the last frame is shifted) or any other random frame to verify if the zone still isolates the feature containing the keypoint. It must also bear in mind that the user must not design too large of a zone to keep computation time reasonable, since every zone is treated as an individual sub-image sequence. The number of keypoints depends on the geometric transformation that will further be used (at least two keypoints are necessary; more is better) and should be distributed as much as possible throughout the reference image to avoid collinearity. The keypoint pixel positions will be defined as $(\kappa_u^{z,c}, \kappa_v^{z,c})$, where $z$ is the sub-image zone index and $c$ is the camera number index.

Figure 2 shows an example of the selection of four different sub-image zones for camera 2 of the Anglet video station. The sizes of the zones vary, but are kept to roughly $250 \times 90$ pixels. It is worthy of mention that it is not advised to choose groins as keypoints, since the corresponding zone will include non-stationary features from the water (such as foam patches) that can be mistakenly identified as keypoints, and can introduce significant errors during the image stabilization method [28]. For the present study, four zones were defined for cameras 1 and 2, and five zones for camera 3. The reference image for each camera was defined as the first frame of each image sequence (1 October, 2013—09:40:00 GMT).

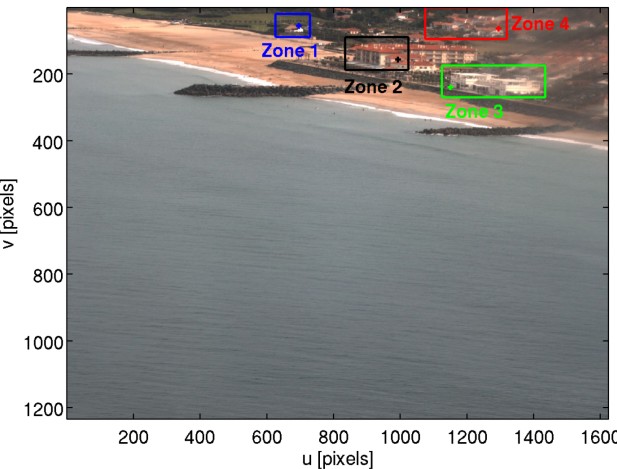

**Figure 2.** Selection of different fixed land regions (zones) with their respective keypoints for camera 2.

## 3.2. Canny Edge Detector

Many popular algorithms automatically recognize, extract, and match keypoints between frames without any a priori knowledge, using corner and blob recognition under similar illumination conditions [50]. Examples of widely used feature detection algorithms are: SIFT (Scale-Invariant Feature Transform; [29,30]), FAST (Features from Accelerated Segment Test; [51]), and SURF (Speeded-Up Robust Features; [31]). Although these algorithms have proven to be very efficient in object recognition applications, their performance is only robust against small brightness changes between frames. This does not represent an inconvenience if the image sequence to be stabilized is sampled at a high-frequency

frame rate over a small period of time (e.g., UAV survey at 2 Hz over 17 min) since the contrast and brightness between frames barely change. However, in long-term coastal monitoring, it is common to work with sub-sampled image sequences (e.g., daily images, as in this study) where consecutive frames usually change substantially in terms of brightness and contrast, owing to different weather and daylight conditions leading the feature detector algorithms to degrade and fail.

The present stabilization method makes use of a Canny edge detector (*CED*; [52]) to enhance and extract the edges that define the boundaries of the primary features within a sub-image zone. Although the *CED* is not insensitive to changes in illumination, it is quite robust at recognizing the same edges between frames even under luminosity variations. The *CED* works in five steps: (a) Smooth the image using a Gaussian filter in order to remove the noise, (b) determine the image intensity gradient in both vertical and horizontal directions, (c) apply a non-maximum suppression to discard all of the points that do not occur at the gradient local maximum, (d) apply a double threshold to preserve certain edges, and (e) track and keep weak edges that are connected to strong edges via an edge threshold (referred to as hysteresis).

### 3.3. Image Sub-Pixel Cross-Correlation and Translation

Before registering and aligning images, a mathematical relationship should be established so that pixel coordinates can be mapped from one image to another. In this sense, a displaced image $g(u', v')$ can be expressed as a coordinate transformation $(T)$ of a reference image $f(u, v)$. A 2-D affine transformation that considers only translation is valid to apply if the size, shape, and orientation of the displaced image remain the same as those of the reference image. This mapping can be represented as a matrix multiplication $(g = Tf)$ using homogeneous coordinates [21]:

$$\begin{bmatrix} u' \\ v' \\ 1 \end{bmatrix} = \begin{bmatrix} 1 & 0 & \Delta u \\ 0 & 1 & \Delta v \\ 0 & 0 & 1 \end{bmatrix} \begin{bmatrix} u \\ v \\ 1 \end{bmatrix} = \begin{bmatrix} u + \Delta u \\ v + \Delta v \\ 1 \end{bmatrix}, \tag{1}$$

where $\Delta u$ and $\Delta v$ denote the two-dimensional displacement (i.e., pixel shift) of $g(u', v')$ with respect to $f(u, v)$.

The 2-D pixel rigid translation $(\Delta u, \Delta v)$ is obtained using the optimized cross-correlation algorithm of Guizar-Sicairos et al. [53] (also referred to as single-step discrete Fourier transform algorithm). This algorithm allows registering the image displacement within a user-specified fraction of a pixel. The algorithm starts by up-sampling two times the resolution of the image dimensions $(M$ and $N)$ and applying a discrete fast Fourier transform (DFT) over all image points $(u, v)$ to get an initial estimate of the location of the cross-correlation peak. The cross-correlation $(CC_{fg})$ of $f(u, v)$ and $g(u', v')$ can be expressed as follows:

$$\begin{aligned} CC_{fg}(\Delta u, \Delta v) &= \sum_{u,v} f(u, v) g^*(u' - \Delta u, v' - \Delta v) \\ &= \sum_{m,n} F(m, n) G^*(m, n) \exp\left[i2\pi \left(\frac{m\Delta u}{M} + \frac{n\Delta v}{N}\right)\right], \end{aligned} \tag{2}$$

where $(*)$ denotes complex conjugation and uppercase letters represent the DFT of the images, as given by:

$$F(m, n) = \sum_{u,v} \frac{f(u, v)}{\sqrt{MN}} \exp\left[-i2\pi \left(\frac{mu}{M} + \frac{nv}{N}\right)\right]. \tag{3}$$

For a higher up-sampling factor, instead of computing all of the up-sampled points of the array, only a small neighborhood around the initial peak estimate is up-sampled using a matrix-multiply discrete Fourier transform to refine the peak location. This dramatically reduces computation cost without losing accuracy. A detailed description of the algorithm is beyond the scope of the present

contribution and can be found in [53,54]. In addition, a MATLAB code version of the algorithm is freely available and can be found in the MATLAB MathWorks File Exchange [55].

Once the $\Delta u$ and $\Delta v$ corresponding to the cross-correlation peak are found, the displaced image can be relocated to the reference image position through an inverse transformation by mapping the pixel coordinates back to the reference image ($f = T^{-1}g$):

$$
\begin{bmatrix} u \\ v \\ 1 \end{bmatrix} = \begin{bmatrix} 1 & 0 & -\Delta u \\ 0 & 1 & -\Delta v \\ 0 & 0 & 1 \end{bmatrix} \begin{bmatrix} u' \\ v' \\ 1 \end{bmatrix} = \begin{bmatrix} u' - \Delta u \\ v' - \Delta v \\ 1 \end{bmatrix}. \tag{4}
$$

As the image shift has sub-pixel accuracy, pixel values at fractional coordinates need to be retrieved. This is achieved by interpolating the surrounding pixels onto the reference image grid in order to compute the new intensities.

2-D rigid image translation is properly valid only for a perpendicularly oriented camera. For an oblique view, the 2-D pixel rigid translation between frames can be interpreted as projections of the camera movement in the azimuth and tilt direction ($\Delta u = \Delta Azimuth, \Delta v = \Delta Tilt$; [1,22,23]). Given the changes in perspective, the 2-D shifts between frames will also vary depending on the sub-region of the image. For the present method, 2-D shifts are computed only with the purpose of matching keypoints distributed in different zones of the image. The stabilization of the whole image, on the other hand, will be performed using these keypoints with another type of geometric transformation with more degrees of freedom to account for the image deformation produced by the relative angle variation of the camera.

At this stage, the keypoints and their corresponding surrounding zones have been defined and filtered through the *CED* in order to extract the edges of the features and improve the performance of the cross-correlation algorithm. The next step consists of using these features along consecutive frames to estimate, via sub-pixel cross-correlation, the two-dimensional displacement of each zone and camera ($\Delta u_{j,r}^{z,c}, \Delta v_{j,r}^{z,c}$), where $j$ is the frame number index and $r$ is the reference frame used for the computation of the 2-D sub-pixel shift. Figure 3 shows an example of keypoint matching between a pair of sub-images from zone 1 ($z = 1$) of camera 2 ($c = 2$). First, the *CED* is applied to both sub-images: The reference sub-image ($r = 1$; 1 October, 2013—09:40:00 GMT) and the displaced sub-image ($j = 1301$; 6 Feburary, 2018—09:15:00 GMT). The sub-pixel shift between frames is subsequently computed using an up-sampling factor of 20: $\Delta u_{1301,1}^{1,2} = -2$ pixels, $\Delta v_{1301,1}^{1,2} = -29.2$ pixels. Finally, the shift is used to translate the displaced sub-image to match with the keypoint of the reference sub-image represented by the green cross ($\kappa_u^{1,2} = \kappa_{u'}^{1,2} - \Delta u_{1301,1}^{1,2}, \kappa_v^{1,2} = \kappa_{v'}^{1,2} - \Delta v_{1301,1}^{1,2}$) in Figure 3.

A semi-automatic method was developed to perform the same steps as shown in Figure 3 for a given image sequence. Figure 4 illustrates a single iteration of this procedure. The goal is to compute the pixel shift of every frame with respect to the reference image ($\Delta u_{j,r}^{z,c}, \Delta v_{j,r}^{z,c}$) in order to retrieve the keypoints' displaced positions ($\kappa_{u'_j}^{z,c}, \kappa_{v'_j}^{z,c}$). The routine works under a loop that iterates over all the frames of a given zone and camera. At each iteration, the algorithm automatically applies a *CED* to the sub-image zone and computes the pixel shift of the current frame with respect to the reference frame (case 1). An additional condition is used to assess the congruence of the estimated pixel shift. The condition assumes that the pixel shift between consecutive frames is small. A typical diurnal signal due to thermal expansion has a range of approximately two pixels [1], so even if the image sequence is sub-sampled, it is reasonable to stipulate that the pixel shift of a frame with respect to the previous one should not be larger than 10 pixels, in either the $u$ or $v$ direction. In the case that the condition is not fulfilled, there is no certain way to discern if the estimated pixel shift is anomalously large because the cross-correlation algorithm has been fooled or if the displacement is actually true and resulted from other factors (e.g., human intervention, strong wind, etc.). The *CED* plays a critical role in decreasing the effects of brightness and contrast changes between frames by just retrieving the edges; however, the cross-correlation algorithm is still likely to fail if some edge patterns of the same keypoint do not match, or even worse, if patterns resemble each other mistakenly while corresponding to different

real-world patterns. To verify that the two frames are as similar as possible, a second option (case 2) is proposed to estimate $\Delta u_{j,r}^{z,c}$ and $\Delta v_{j,r}^{z,c}$ without directly computing the pixel shift with respect to the reference frame. Instead, the pixel shift is computed with respect to the previous stabilized frame: $f(u_{j-1}^{z,c}, v_{j-1}^{z,c}) = g(u'^{z,c}_{j-1} - \Delta u^{z,c}_{j-1,r}, v'^{z,c}_{j-1} - \Delta v^{z,c}_{j-1,r})$, which in turn is also relative to the reference frame. If, despite the above two cases, the pixel shift remains greater than 10 pixels, the algorithm enters into a semi-automatic mode and offers the user four different options:

- Keep the pixel shift computed by using the reference frame (case 1).
- Keep the pixel shift computed by using the previous stabilized frame as the reference (case 2).
- Manually select the keypoint position to compute the pixel shift (case 3).
- Discard the frame (case 4).

An additional visual representation (similar to Figure 3) shows the user how the stabilized frame would look if the case 1 or case 2 pixel shift were used. If neither option is appropriate, the user has the possibility to manually select the keypoint position $(\kappa_{u'_j}^{z,c}, \kappa_{v'_j}^{z,c})$ in the current frame in order to compute the corresponding pixel shift (case 3). Alternatively, the frame can present serious contamination in the form of sun glint, raindrops, or fog, in which case the frame can be discarded (case 4). Finally, the pixel shift is stored at each iteration so that all of the keypoints of a given zone and camera can be matched against themselves.

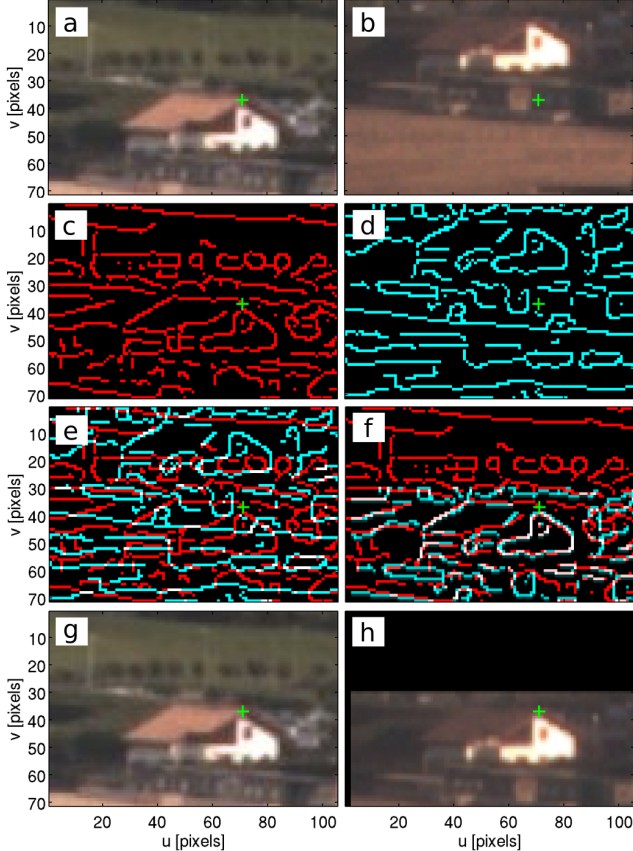

**Figure 3.** Keypoint matching between a pair of sub-images corresponding to zone 1 of camera 2. (**a**) Reference sub-image: 1 October, 2013—09:40:00 GMT; (**b**) unstabilized sub-image: 6 February, 2018—09:15:00 GMT; (**c**) reference sub-image after applying the Canny edge detector (*CED*); (**d**) unstabilized sub-image after applying the *CED*; (**e**) reference and unstabilized sub-image overlapped; (**f**) sub-image translation after computing azimuth and tilt pixel shift; (**g**) same figure as (**a**); (**h**) stabilized sub-image: 6 February, 2018—09:15:00 GMT. The green cross represents the static keypoint that should match between frames.

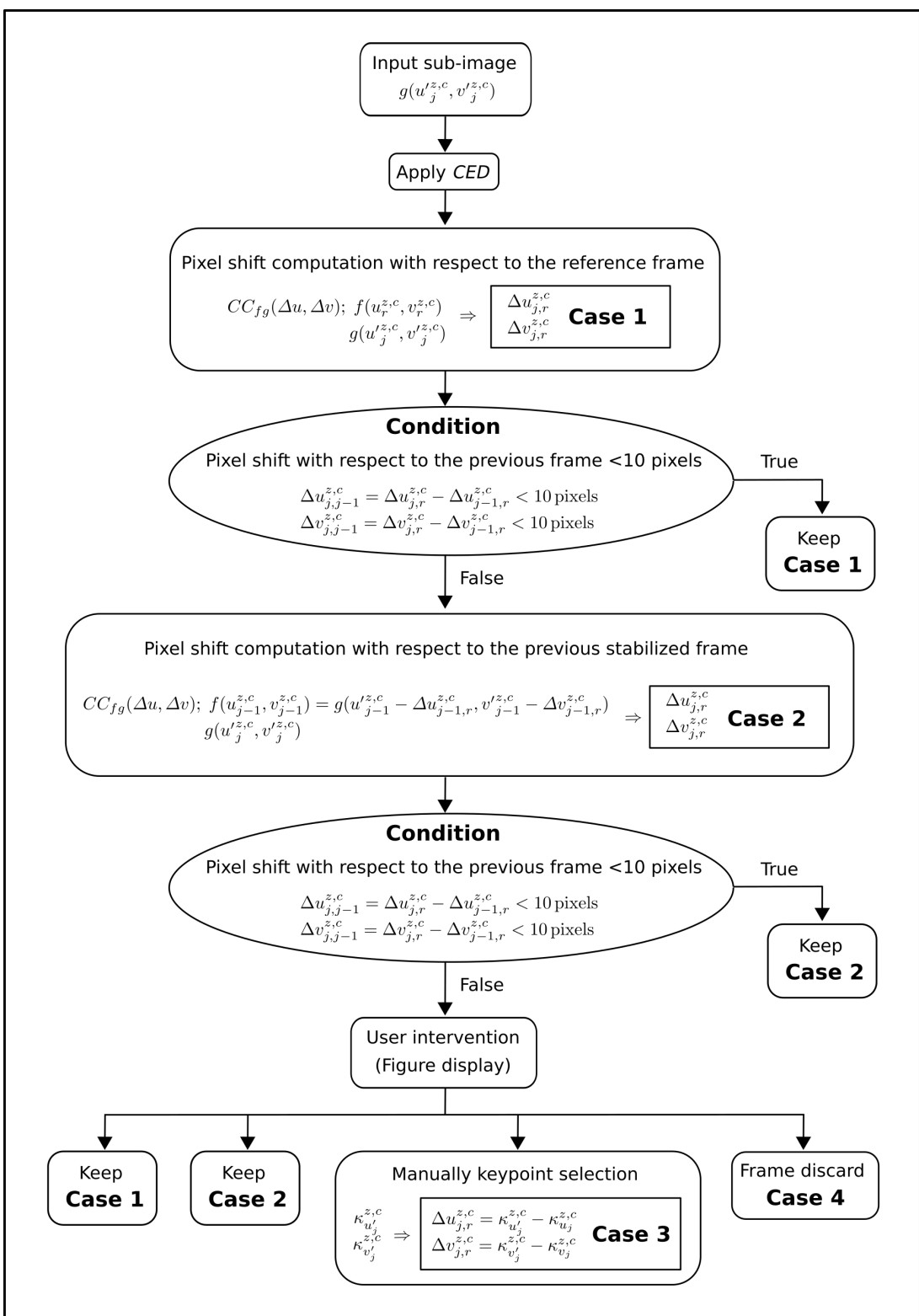

**Figure 4.** Automatization steps to stabilize a sub-image zone of an image sequence by estimating the 2-D pixel shift with respect to a reference frame.

### 3.4. Geometric Transformation

The number of corresponding keypoints identified between images establishes the type of geometric transformation that can be defined to map and relocate the displaced image points onto a reference image. A geometric transformation is constrained by the degrees of freedom (DOF). At least

one corresponding keypoint is necessary to estimate the 2-D rigid translation motion between frames (two DOF). Two pairs of non-collinear keypoints are enough to solve the four parameters of a similarity transformation:

$$\begin{bmatrix} u' \\ v' \\ 1 \end{bmatrix} = \begin{bmatrix} s\cos\theta & -s\sin\theta & \Delta u \\ s\sin\theta & s\cos\theta & \Delta v \\ 0 & 0 & 1 \end{bmatrix} \begin{bmatrix} u \\ v \\ 1 \end{bmatrix} = \begin{bmatrix} us\cos\theta - vs\sin\theta + \Delta u \\ us\sin\theta + vs\cos\theta + \Delta v \\ 1 \end{bmatrix}, \tag{5}$$

defined as a linear combination of translation ($\Delta u, \Delta v$), rotation ($\theta$), and a scale factor ($s$). The rotation is defined as a counter-clockwise rotating angle with respect to the reference frame and can often be interpreted as the camera angle movement on its roll axis. The similarity transformation lacks the ability to represent a true three-dimensional motion model; however, it can adequately estimate motion between frames if the camera movement or the scene are constrained to certain conditions [56]. Jin et al. [57] stated that if the translations of the camera are zero and the relative angle variation of the camera between frames is very small (e.g., less than 5°), the effect of the depth of scene points can be neglected. This means that camera roll is the main factor of image rotation ($\theta = \Delta Roll$), and small camera azimuth and tilt variations can be represented by 2-D translations of the image ($\Delta u = \Delta Azimuth, \Delta v = \Delta Tilt$). On the other hand, if image points are of the same depth, or the depth difference among image points is much less than the average depth (i.e., the camera is far away from the object), small camera translation will mainly cause homogeneous scaling and translation of 2-D images [58]. The latter assumptions are valid for the case of fixed cameras as long as the viewing angles shift slightly. It is important to note that camera orientation always goes in the opposite direction from the relative frame displacements with respect to the reference image (Figure 5).

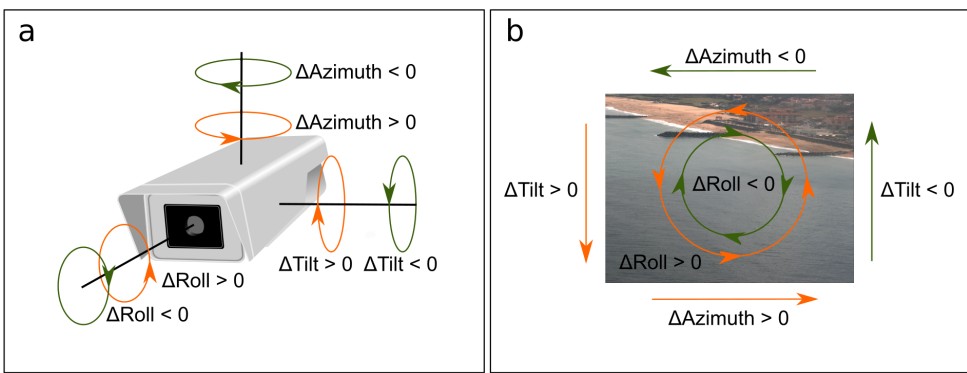

**Figure 5.** (**a**) Camera angle movements according to (**b**) azimuth, tilt, and roll deviations with respect to a reference image.

Working with a similarity transformation has the advantage of being fast and simple in terms of computational complexity [59]. The similarity motion parameters can be estimated over an iterative process using the keypoint correspondences of all sub-images' zones for every frame of a given camera. These parameters allow the performing of a global transformation to warp and stabilize the whole image sequence.

### 3.5. Image Geo-Rectification

After all the images are correctly stabilized, the coordinates of the 2-D images ($u, v$) must be transformed to 3-D real-world coordinates ($x, y, z$) through a photogrammetric transformation. The determination of the transformation is called camera calibration and involves two sets of parameters: Extrinsic and intrinsic parameters. The extrinsic parameters contain the information related to the position and orientation of the camera (the three coordinates of the camera location and the three rotation angles) and the intrinsic parameters comprise the physical characteristics of the lens of the camera (the image center coordinates, the effective focal length, and the scale factors). Direct linear

transformation (DLT) developed by Abdel-Aziz and Karara [60] is perhaps the most commonly used camera calibration method because it does not require initial knowledge of the extrinsic and intrinsic parameters [34]. These parameters are implicit in the 11 transformation parameters; however, nonlinear effects such as radial distortion are not taken into account and can lead to coupling errors between parameters, affecting the camera calibration accuracy [61]. The DLT method represents a closed-form solution requiring non-coplanar real-world surveyed GCPs with corresponding image coordinates. Each GCP generates two linear equations that can be expressed as the following matrix multiplication:

$$
\begin{pmatrix}
x & y & z & 1 & 0 & 0 & 0 & 0 & -ux & -uy & -uz \\
0 & 0 & 0 & 0 & x & y & z & 1 & -vx & -vy & -vz
\end{pmatrix} L =
\begin{pmatrix} u \\ v \end{pmatrix},
\tag{6}
$$

where $L = (L_1, L_2, ..., L_{11})^T$ are the transformation parameters. At least six GCPs are necessary to build an overdetermined set of linear equations to estimate the 11 transformation parameters using a least square method. The limitation of working with one single camera, or many cameras that are not looking at the same object, is that one coordinate must be constrained to a fixed value to allow inverse mapping. When studying the nearshore zone, the vertical coordinate ($z$) is usually assigned to be equal to the instant sea level. However, if the topography is known and the swash zone is the subject of study (e.g., wave runup), an iterative rectification method using the beach topography can be applied to reduce rectification positioning errors, as described by Blenkinsopp et al. [62].

Another popular calibration method is the one proposed by Holland et al. [20]. In their method, they solve the five intrinsic parameters in the lab and solve the six remaining extrinsic parameters using at least three GCPs and a standard nonlinear solver [25,26]. The intrinsic parameters can be estimated using the Caltech camera calibration toolbox for Matlab [63] prior to camera installation, in order to remove lens distortion. Although there are many well-documented camera calibration methods [64–67], deciding which one to use is up to the user's convenience.

For this particular paper, DLT was used to generate the plan-view (geo-rectified) images, considering 8–10 GCPs per camera. Images were merged and projected onto a horizontal plane with a grid size of $1 \times 1$ m and elevation equal to mean high water neaps ($z_{MHWN}$ = 0.86 m). In addition, non-stabilized and stabilized geo-rectified images were compared to estimate the induced real-world positioning error due to camera movement.

## 4. Results

### 4.1. Keypoint Tracking

The stabilization method was applied to five years of daily images collected from the permanent three-camera video system located at Anglet Beach. Figure 6 illustrates estimates of horizontal (azimuth) and vertical (tilt) displacements of the four keypoints distributed at different sub-image regions of camera 2 (see Figure 2). Azimuth and tilt deviations were expressed as pixel differences from the initial image position (1 October, 2013—09:40:00 GMT). The results show a high variability in the position of the four keypoints, indicating that the camera moved throughout the study period. Deviations of azimuth and tilt varied respectively from −10 to 8 pixels and −33 to 2 pixels. The standard deviation of the detrended pixel displacement, which is a measure of the magnitude of camera movements, was 2 and 5 pixels in the horizontal and vertical directions, respectively. The camera oblique view generated an approximately constant offset (0.2 pixels horizontal; 0.4 pixels vertical) in the displacement between zones. An annual motion signal ($\approx$4 pixels amplitude) together with a quasi-steady trend (1.6 pixels/year horizontal; −3.4 pixels/year vertical) can be identified, suggesting that the orientation of camera 2 gradually moved towards the southwestern direction.

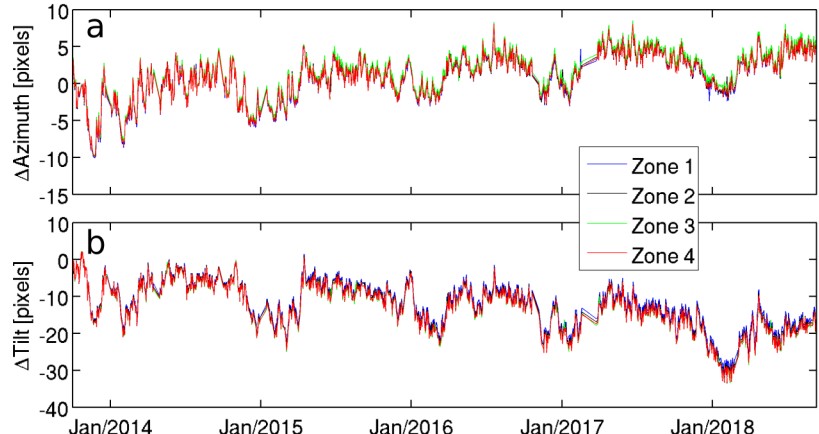

**Figure 6.** Daily (**a**) azimuth and (**b**) tilt displacements of the different sub-image zones in camera 2, expressed in terms of pixel shift. The deviations are computed with respect to the reference image: 1 October, 2013—09:40:00 GMT.

## 4.2. Camera Movement Correction

Figure 7 shows the evolution of the transformation parameters (translation, rotation, and scaling) derived from a Euclidean (i.e., Similarity) transformation for the three-camera video system. All cameras showed motion throughout the study period on a wide range of time scales. The movement amplitudes of cameras 1, 2, and 3 reached approximately 5.6, 10.4, and 5.7 pixels in azimuth and 9.5, 30.7, and 15.1 pixels in tilt, respectively. The standard deviations of detrended azimuth for the three cameras were 1.5, 2.5, and 1.4 pixels with trends of −0.8, 1.6, and −0.4 pixels/year. Respectively, the standard deviations of detrended tilt were 2.9, 4.7, and 1.8 pixels, with trends of 0.8, −2.9, and −2.1 pixels/year. An annual signal of about 0.6 (camera 1), 2.3 (camera 2), and 0.4 pixels (camera 3) in azimuth and 1.9 (camera 1), 4.2 (camera 2), and 1.8 pixels (camera 3) in tilt deviation were apparent on the total record. Moreover, the three cameras presented variations in roll angle up to 0.4°, 0.2°, and 0.1° with a standard deviation of 0.05°, 0.03°, and 0.03° for cameras 1, 2, and 3, respectively. Cameras 1 and 2 exhibited an increasing counter-clockwise trend in roll angle (0.08°/year and 0.03°/year, respectively) over the five-year time series. An overall enlargement was notable after 2015, particularly for camera 1. The scale factor (*s*) increased the image size of camera 1 by up to 10% and the image size of camera 2 by 3%. Camera 3 presented no significant changes in image scale (maximum of 0.6%).

The transformation parameters were used to warp and align the whole image sequence with the reference frame. Figure 8 shows the long-term averaged video (spanning the complete time series) using raw video frames and corrected frames from camera 2. The raw input mean is blurry (Figure 8a), indicating significant image shifts induced by the camera movement. The mean of the corrected frames (Figure 8b) shows the efficacy of the stabilization method by preserving the same image sharpness (of the fixed features) as the reference frame.

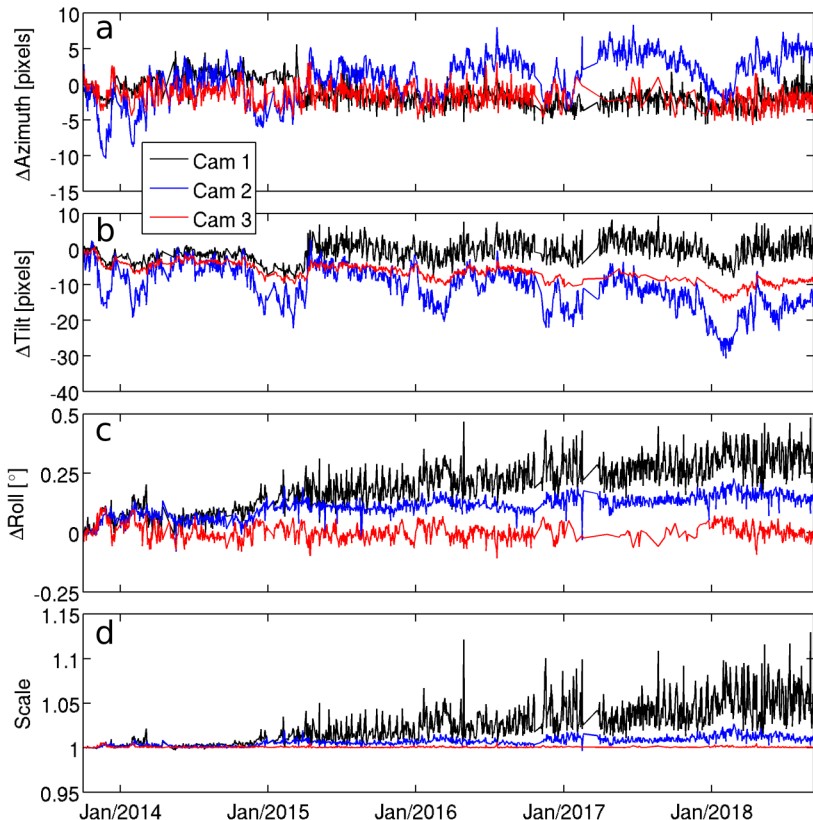

**Figure 7.** (**a**) Azimuth, (**b**) tilt, (**c**) roll, and (**d**) scale parameters of the similarity transformation matrix computed for the three cameras relative to the reference image: 1 October, 2013—09:40:00 GMT.

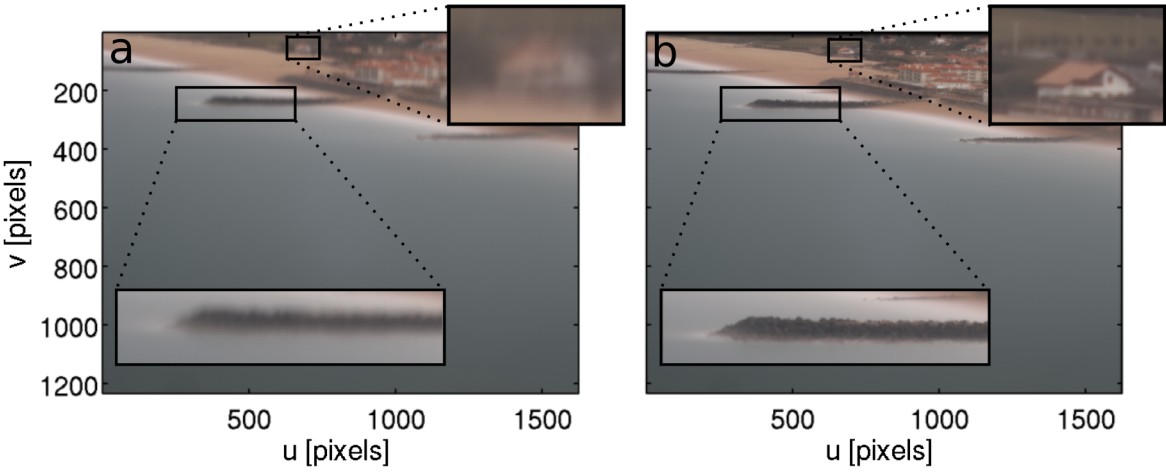

**Figure 8.** Time series average (spanning 1 October 2013–9 September 2018) of all of the (**a**) unstabilized frames and (**b**) stabilized frames of camera 2.

*4.3. Geo-Rectification Error*

Camera movements result not only in image deformation but also in geo-rectification errors. Figure 9 provides an example of positioning error due to camera motion. A comparison between stabilized and unstabilized geo-rectified images for a specific date is presented together with a spatial description of the error, to highlight the impact of camera viewing angle deviations. Figure 9a shows the reference frame (1 October, 2013—09:40:00 GMT) plain-view image in real-world

coordinates obtained after geo-rectification of the time-exposure images collected from the three-camera video system. Figure 9b,c show geo-rectification of a subsequent image when a large camera viewing angle deviation was present (6 Feburary, 2018—09:15:00 GMT; same date chosen as for Figure 3). Figure 9b considers stabilization of the time-exposure images before geo-rectification, while Figure 9c does not take it into account. Figure 9d presents the associated geo-rectification error ($\sqrt{\text{alongshore error}^2 + \text{cross-shore error}^2}$) of Figure 9c produced by camera movement. This error depends on the grazing angle and lens properties, and typically increases with increasing distance from the camera [24]. The comparison between images highlights the differences in position, size, and shape of fixed objects when stabilization is omitted. For this particular date, horizontal errors exceed 400 m (400 m in the alongshore and 100 m in the cross-shore directions). For instance, a displacement of the groins is readily apparent, with the alongshore location of the right-hand groin varying from 1750 to 2000 m. Furthermore, the whole beach area appeared to shrink approximately 50 m in contrast to the stabilized geo-rectified image. The geo-rectified image that considered stabilization kept the groins well aligned with respect to their original positions, with all of the fixed objects retaining their initial sizes and shapes.

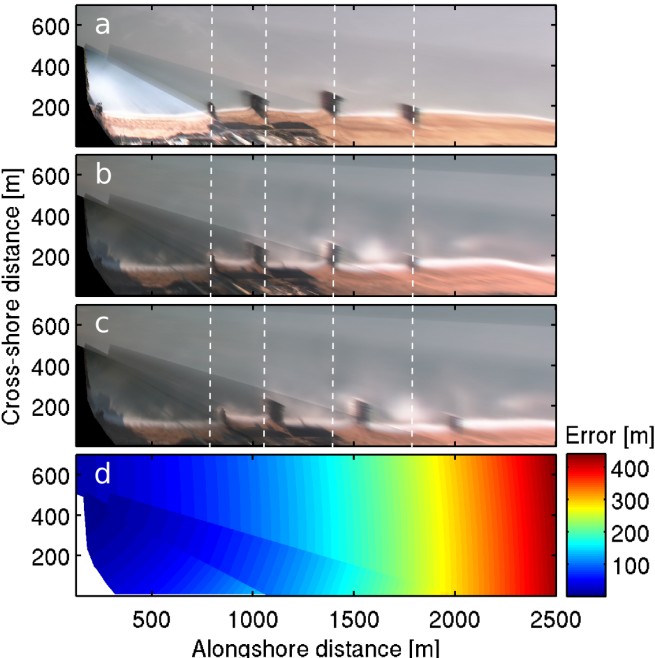

**Figure 9.** Anglet Beach geo-rectified time-exposure images highlighting the impact of camera viewing angle deviation. (**a**) Reference geo-rectified image: 1 October, 2013—09:40:00 GMT; (**b**) stabilized geo-rectified image: 6 February, 2018—09:15:00 GMT; (**c**) unstabilized geo-rectified image: 6 February, 2018—09:15:00 GMT; (**d**) positioning error due to camera movement for 6 February, 2018—09:15:00 GMT. White dashed lines indicate groins' original positions.

## 5. Discussion

The semi-automatic stabilization method was applied to five years of daily time-exposure images collected from three synchronized cameras located at Anglet Beach in southwestern France. For keypoint matching, approximately 1500 frames were processed for each sub-image zone, requiring an average processing time of between 1 and 4 s per frame on a standard commercial personal computer (4th generation Core i5-4690, 3.5 GHz, Dell OptiPlex 9020). The distribution of cases used to compute the 2-D pixel shift displacement for each keypoint (Figure 4) gives an estimation of the performance of the algorithm in terms of how much user assistance was required. For example, just for sub-image zone 1 of camera 2, 97% of the pixel shifts were computed using case 1, 0.92% were computed using

case 2, 0.26% using case 3, and 1.7% using case 4. This means that four frames corresponding to that zone required manual pixel shift computation and 26 frames were manually discarded. The overall distribution of cases used for all cameras and all sub-image zones was: 90.42% (case 1), 1.38% (case 2), 4.91% (case 3), and 3.28% (case 4). It is important to note that some matching cases corresponding to case 1 and case 2 also required user visual control. User confirmation for those cases was necessary when the automatically computed pixel shift between consecutive frames was larger than the threshold (10 pixels) but was still correct. Much of the user intervention essentially consisted of discarding low-quality frames. For future work, a possible improvement could be achieved by pre-selecting frames or exploring this issue further by developing a separate procedure to automatically keep/discard frames prior to image stabilization.

The use of the *CED* is crucial for a robust performance of the cross-correlation algorithm when differences in contrast, brightness, and illumination conditions between frames are present. Figure 10 shows the effect of the *CED* on the stabilization with respect to a reference sub-image (Figure 10a,e,i) under three representative cases of illumination conditions: Under shiny (overexposed image; Figure 10b), cloudy (high-contrast image; Figure 10f), and foggy (low-contrast image; Figure 10j) weather conditions. The cross-correlation between sub-images without *CED* determines the similarity between frames based on their color/grayscale pixel intensity. This means that the performance of the cross-correlation not only depends on the features present in the frame, but also on changes in light conditions between frames. Large changes can lead to errors, as shown in Figure 10c,g,k. On the other hand, when using the *CED*, pixel intensities are converted to only two possible values: 1 if edges that define the boundaries of the features are detected, and 0 if not. The efficiency of the cross-correlation in combination with the *CED* relies upon the fact that the amount of data to be processed is reduced. Moreover, the cross-correlation algorithm, together with the *CED*, estimates the similarity between frames based only on the boundaries and edges detected, regardless of contrast and brightness variations between frames, as long as the features are still visible and easily recognizable by their shapes and forms (Figure 10d,h,l).

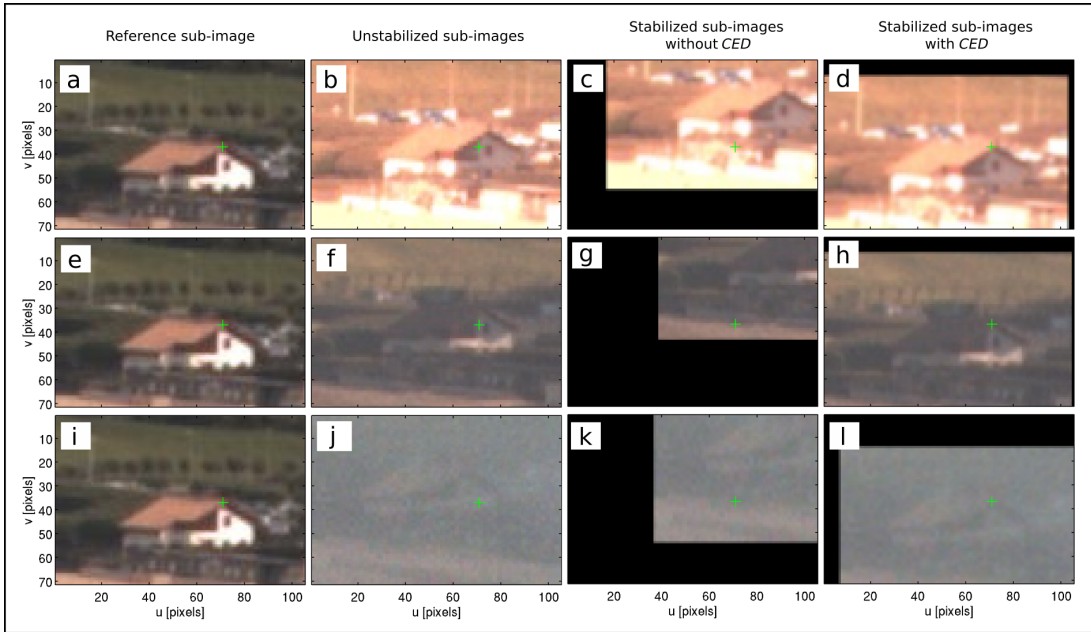

**Figure 10.** Sub-image stabilization of zone 1 of camera 2 under different illumination conditions. (**a,e,i**) Reference sub-image: 1 October, 2013—09:40:00 GMT; (**b,f,j**) unstabilized sub-images under shiny, cloudy, and foggy weather conditions; cross-correlation between the reference sub-image and the unstabilized sub-images with (**d,h,l**) and without (**c,g,k**) using the *CED*. The green cross represents the static keypoint that should match between frames.

The performance of the cross-correlation algorithm (together with the *CED*) depends strongly on the land-region zone selected; therefore, the sub-image region selection is the most important step of the method. It is advised to define a large enough window to allow inter-frame movement not only for the keypoint, but also for the whole feature (e.g., building structure, billboard, road, etc.) to guarantee better results and reduce user assistance during the pixel shift computation process as much as possible. The compromise of having a larger zone-size is just related to the computation time that it will take for the cross-correlation algorithm to work on each frame. However, it is more important to be careful not to choose a very small zone where the feature (including the keypoint) might drift out of view; in this case, this would lead the cross-correlation algorithm to fail. A limitation of the method is that it requires the presence of fixed features in the camera view field. However, for fixed video systems where cameras' viewing angles usually shift slightly (e.g., less than 5°) and cameras' translational movements are small, a similarity transformation (four DOF) can be performed to stabilize the images with a minimum of two non-collinear pairs of keypoints between frames. The advantage of this approach, besides requiring fewer keypoints, is that it is computationally efficient and robust enough to achieve good results. Nevertheless, registration reliability and accuracy can be increased by selecting more landmarks in the image. Having more spread keypoints in the image reduces foreground–background bias, and also allows the possibility of implementing other more complex types of geometric transformations with higher degrees of freedom [21], such as a 2-D affine (six DOF) or planar nonlinear homography (eight DOF) transformation, to remove perspective deformation introduced by the camera oblique view (really necessary for UAVs). The method has been tested on UAV flights (data not shown), showing that the performance largely improves (with 99% matches for case 1 and case 2) when using higher frequency frame rates (>1 Hz) over smaller periods of time (<20 min), although other geometric transformations with higher degrees of freedom are necessary, due to the fact that camera translation movements become significant.

The time evolution of the transformation parameters shows that camera movements occur on a wide range of timescales (see Figure 7). The annual signal in azimuth and tilt deviation can potentially be attributed to the sun's position and thermal expansion fluctuations [24]. It is important to note that all cameras were installed inside the lighthouse, mounted in a wooden structure, and isolated by an acrylic glass from the outside elements, so wind is not expected to be a source of camera motion for the present study. While movements were likely occurring for all cameras, the effects of azimuth and tilt motion were most notable for camera 2 (which had the longest focal length; 25 mm lens) and less evident for camera 3 (wide-angle 8 mm lens). This is in line with Pearre and Puleo [22], who previously showed that the longer the focal length (e.g., 50 mm), the smaller the angle of view, and the larger the sensitivity to causing significant changes in image location, even for small changes in the tilt of the camera. Moreover, small changes in perspective were found to modify the sizes of the objects (uniform scaling) when objects were far away from the camera, as shown in Figure 7d for camera 1. On the other hand, the quasi-steady counter-clockwise trend observed in roll angle (Figure 7c) suggests that the wooden structure where the cameras are mounted is gradually arcing with time, although this cannot be verified. Previous studies have shown that small camera viewing angle deviations can induce significant changes in image location and, in turn, introduce large geo-rectification errors [22,24,26]. However, besides camera variation angles, the related error is also modulated with the pixel footprint, which depends on the distance from the camera and the lens properties. For example, even though camera 2 presented the largest deviations in azimuth and tilt (see Figure 6), its associated geo-rectification error was slightly lower with respect to the other cameras for the same distances (see Figure 9d). This result might be explained by the fact that camera 2 has a higher pixel resolution and, hence, a lower pixel footprint that counteracts the pixels' induced real-world location error. Nevertheless, the evidence still points out that geo-rectification errors induced by camera movement can become significant and should not be neglected. In an attempt to demonstrate the impact of this error (in perhaps the most common coastal video application), 2.5 km of shoreline were manually digitized for the outstanding winter period of 2013/2014 [68–70], using non-stabilized

and stabilized time-exposure geo-rectified images to estimate the real-world horizontal positioning error due to camera movement (Figure 11). The mean shoreline position (intersection of wet and dry parts of the beach [4]) was defined as the alongshore-averaged shoreline position. The alongshore standard deviation of the shoreline position was also computed to give a measure of the alongshore variability of the cross-shore position. During the winter period of 2013/2014, high-energy wave conditions ($H_s$ > 5 m) drove important changes in Anglet Beach shoreline dynamics [5]. Part of the seasonal erosion cycle is captured in Figure 11, where the shoreline position varied within a range of 30 m. The mean shoreline apparent position was also affected by an approximately 10–20 m bias, showing that incorrect tilt may bias positions of features by condensing or stretching the image in the cross-shore direction. This result highlights that overlooking camera movement can result in strongly under- or over-estimation of shoreline response to extreme events.

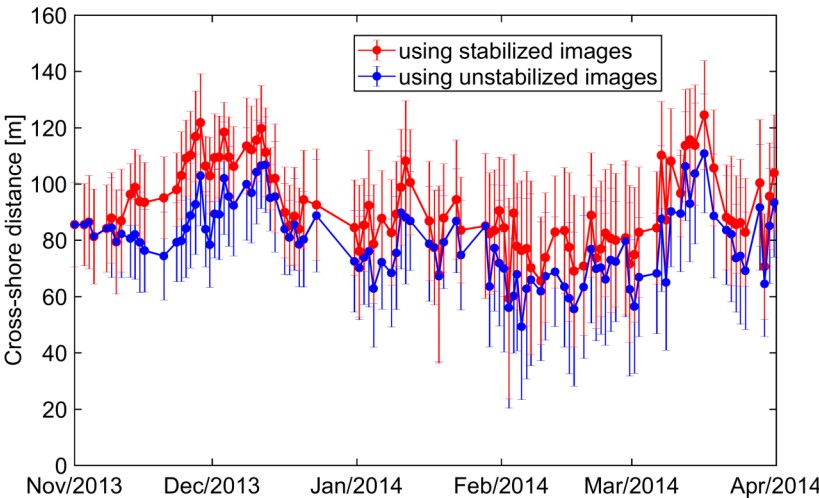

**Figure 11.** Daily alongshore averaged shoreline position extracted from stabilized (red dots) and unstabilized (blue dots) geo-rectified images for the outstanding winter period of 2013/2014. The alongshore standard deviation of the cross-shore position is indicated by the vertical error bars.

## 6. Conclusions

In this paper, we developed an efficient semi-automatic method to remove unwanted camera movement after video acquisition. The method consisted of defining and tracking a few fixed feature points between consecutive frames using the Guizar-Sicairos et al. [54] sub-pixel cross-correlation algorithm together with a *CED* [52]. The use of the *CED* greatly improved the performance of the cross-correlation algorithm by making it more robust against contrast and luminosity brightness variations. The tracked features allowed the computation of the parameters of a similarity transformation (translation, rotation, and scale) in order to estimate the motion between frames and compensate for it. For the keypoint matching, the method worked under a scheme of four cases. The algorithm was capable of automatically computing the 2-D sub-pixel shift of a keypoint with respect to an initial position, as long as the displacements between consecutive frames remained smaller than 10 pixels. Otherwise, user input was required to discern if the calculated pixel shift was legitimate or if the frame had to be discarded. The results showed that the semi-automatic method is able to process at least 90% of the frames without user assistance. However, future work should examine the possibility to automatically discard low-quality images acquired under adverse rainy or foggy conditions, in order to reduce user intervention. Image stabilization is a fundamental post-processing step that should always be performed in coastal imaging applications to increase the accuracy of video-derived products, such as shoreline/sandbar positions, wave celerity, and depth estimates. The framework presented here opens new perspectives and appears as a promising tool for other coastal imaging applications, such as removal of undesired high-frequency jitters from UAVs.

**Author Contributions:** I.R.-P. wrote the original draft of the manuscript. B.C., V.M., and D.M. reviewed and edited the manuscript. I.R.-P., B.C., and V.M. contributed to the research concept, analysis, and interpretation of data. V.M. provided his expertise in video-image processing and aided with the design of the stabilization method. D.M. provided the video data. I.R.-P. developed the stabilization method. All authors have read and agreed to the published version of the manuscript.

**Funding:** This research received external funding by CONACyT (México) through the PhD scholarship of I.R.-P. (grant 540839).

**Acknowledgments:** The authors are thankful to E. Bergsma and C. Bouvier for their valuable comments on the method, and would also like to thank Abril Jiménez for kindly digitizing the shoreline position. Anglet beach is a monitoring site labelled by the Service National d'Observation (SNO) Dynalit (https://www.dynalit.fr). Storage resources for this study were provided by the computing facilities MCIA (Mésocentre de Calcul Intensif Aquitain) of Univ. Bordeaux and Univ. Pau et Pays de l'Adour. The OCA (Observatoire de la Côte Aquitaine) has provided financial support to ensure the maintenance of the video station. The authors would like to thank the three anonymous reviewers for their constructive and positive feedback.

**Conflicts of Interest:** The authors declare no conflict of interest.

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
