# Peer review of "A Simple and Efficient Image Stabilization Method for Coastal Monitoring Video Systems"

_remotesensing, doi:10.3390/rs12010070_

Round 1

Reviewer 1 Report

In the introduction the authors carried out a literary review in the accessible low-cost tool for coastal monitoring . They underline the problem of image stabilization, above all ,in outdoor installations where cameras are directly exposed to the elements, so they present a new method for image stabilization to overcome this limitation.

In the paragraph 2 they present study site and video data.

The site is Angke Beach in the Basque coast in France where a permanent video monitoring station has been operating since September 2013 .Tha station consists of three cameras . They collected every 20 min at 1Hz, from March 2017 the configuration changed and the image collection was every 15 min ay 1HZ durin an average period of sampling of 14 min.

In the paragraph 3 the stabilization method is described, then they present a stabilization method based on feature matching that necessarly requires the presence of at least few recognizable static features distributed in both domensions in the field of view.

The authors underline that the user needs to anticipate and guess the camera pixel shift in order to define the minimum size of the zone the allows for some inter-frame movement pf the keypoint.

But if this isn’t possible?

Then they present the Canny edge detector (CED)to enhance and extract the edges that define the boundaries of the primary features within a sub image zone. Then image sub pixel cross correlation and translation. The goal is to compute the pixel shift of every frame with respect to the reference image in order to retrieve the keypoints displaced positions. Geometric trasformation, then the image geo-rectification are necessary.

The paragraph 4.3 : Fig 9 perhaps may be advisable a better explanation . With 9°, b e c for me isn’t available.

In the conclusion the authors talk about the UAV case study. Perhaps shuold be better esxplain teh case study with UAV photogrammetry?

Reviewer 2 Report

Dear authors,

The authors developed a stabilizing method to remove unnecessary motions in video frames. The method is based on feature-matching and sub-pixel cross-correlation techniques using CED. In results, image geo-rectification was successfully performed. The stabilizing the time series of images are troublesome pre-processing for image analyses. The provided semi-automatic algorithm may be useful for many users of videos and time-lapse cameras for monitoring of various targets.

This manuscript is described clearly, however, I think it needs some revisions.

Major comments,

The main purpose of this study is to provide a robust solution to stabilize an image sequence under varying illumination using CED. And the authors concluded that the use of CED greatly improved the performance of the cross-correlation algorithm. However, the effects of CED were not demonstrated clearly. You need to compare the results between with and without CED, and to discuss the efficiencies in detail. In addition, specific illumination conditions are not described. Please show the results under different contrast, brightness and illumination conditions. Were the methods effective under shinny or foggy weather as well? How much percentages were corresponded for user intervention cases? 97 % of case 1 and 0.92 % of case 2 included both automatic and user intervention processes. Please show us some specific examples of such cases.

Minor comments,

Introduction: Similar stabilizing methods have already been developed for time-lapse camera images to monitor various targets other than coast lines. You can quote additional references.

Line 338: in order to?

Fig 5 Horizontal and vertical linear arrows would be better for motions of azimuth and tilt, respectively, than rotated arrows.

Fig. 6 ΔAzimuth and ΔTilt would be better than Δu and Δv for the names of vertical axes in accordance with Fig 7.

Reviewer 3 Report

The paper proposes an efficient semi-automatic method to remove unwanted camera movement after video acquisition. The paper is basically well written and refers suitable papers. The results of the proposed method are convincing and practically very valuable. I appreciate the authors' efforts to make the paper very attractive. So my decision is to accept.

Author Response

We thank Reviewer #3 for their positive feedback and support for publication.